# Comparison Approach for Identifying Missed Invasive Fungal Infections in Formalin-Fixed, Paraffin-Embedded Autopsy Specimens

**DOI:** 10.3390/jof8040337

**Published:** 2022-03-24

**Authors:** Sota Sadamoto, Yurika Mitsui, Yasuhiro Nihonyanagi, Kazuki Amemiya, Minoru Shinozaki, Somay Yamagata Murayama, Masahiro Abe, Takashi Umeyama, Naobumi Tochigi, Yoshitsugu Miyazaki, Kazutoshi Shibuya

**Affiliations:** 1Department of Surgical Pathology, Toho University School of Medicine, Tokyo 143-8541, Japan; souta.sadamoto@med.toho-u.ac.jp (S.S.); nihonyanagi@sakura.med.toho-u.ac.jp (Y.N.); kazuki.amemiya@med.toho-u.ac.jp (K.A.); ushino@med.toho-u.ac.jp (M.S.); kaz@med.toho-u.ac.jp (K.S.); 2Department of Fungal Infection, National Institute of infectious Diseases, Tokyo 162-8640, Japan; smurayam@niid.go.jp (S.Y.M.); masa-a@niid.go.jp (M.A.); umeyama@niid.go.jp (T.U.); ym46@niid.go.jp (Y.M.); 3Department of Hematology and Oncology, Toho University School of Medicine, Tokyo 143-8541, Japan; yurika.mitsui@med.toho-u.ac.jp

**Keywords:** formalin-fixed paraffin-embedded (FFPE), invasive fungal infection (IFI), autopsy, polymerase chain reaction (PCR), in situ hybridization (ISH), immunohistochemistry (IHC), mucormycosis, *Cunninghamella*

## Abstract

Invasive fungal infection (IFI) has a high mortality rate in patients who undergo hematopoietic stem cell transplantation, and it is often confirmed by postmortem dissection. When IFI is initially confirmed after an autopsy, the tissue culture and frozen section are challenging to secure, and in many cases, formalin-fixed, paraffin-embedded (FFPE) samples represent the only modality for identifying fungi. Histopathological diagnosis is a useful method in combination with molecular biological methods that can achieve more precise identification with reproducibility. Meanwhile, polymerase chain reaction (PCR) using fungal-specific primers helps identify fungi from FFPE tissues. Autopsy FFPE specimens have a disadvantage regarding the quality of DNA extracted compared with that of specimens obtained via biopsy or surgery. In the case of mucormycosis diagnosed postmortem histologically, we examined currently available molecular biological methods such as PCR, immunohistochemistry (IHC), and in situ hybridization (ISH) to identify fungi. It is reasonable that PCR with some modification is valuable for identifying fungi in autopsy FFPE specimens. However, PCR does not always correctly identify fungi in autopsy FFPE tissues, and other approaches such as ISH or IHC are worth considering for clarifying the broad classification (such as the genus- or species-level classification).

## 1. Introduction

Culture should be considered the gold standard for diagnosing infectious diseases. However, because of its insensitivity regarding invasive fungal infection (IFI), several diagnostic procedures are usually applied in combination in the clinic, such as direct microscopy using clinical specimens, histopathological examination, diagnostic imaging, and biochemical markers in serum.

Histopathological diagnosis of IFI is an important and frequently used diagnostic method, but it can be challenging to differentiate species with similar morphologies [1,2]. In recent years, several molecular biological approaches such as polymerase chain reaction (PCR) and in situ hybridization (ISH) have been used to identify causative fungi. Recently, the European Organization for Research and Treatment of Cancer and the Mycoses Study Group Education and Research Consortium (EORTC/MSGERC) guidelines suggested that PCR targeting the internal transcribed spacer (ITS) or D1/D2 region of rDNA can be used for species determination in formalin-fixed, paraffin-embedded (FFPE) tissue only when fungal elements are confirmed by histopathology diagnosis [3].

Because analysis using next-generation sequencers has recently become popular in the field of cancer examination, the standardization of quality control of FFPE specimens for biopsy and surgical materials has advanced appropriately [4,5,6]. By contrast, few studies have assessed the quality of FFPE specimens obtained via autopsy, which might have DNA fragmentation because of their long period of formalin fixation [7,8,9]. Some data from cancer research suggested that FFPE samples are associated with decreased gene extraction efficiency mostly because of deterioration over time [10]. These points may also apply to the detection of causative fungi in FFPE samples. Therefore, the results of some previous reports assessing the usefulness of supplemental gene diagnostic methods using FFPE samples for IFI diagnosis suggest that ISH could be performed together with PCR because the amplification procedure of the latter technique could result in false-negative results [7].

Retrospective epidemiological data provided valuable information for diagnosing IFIs such as mucormycosis, which are difficult to diagnose prenatally [11,12,13]. If IFI was not expected premortem, FFPE tissue blocks could be the only available materials for identifying the causative fungi. Even in such situations, molecular diagnostic tools can be applied. However, compared with PCR, other adjunctive diagnostic methods such as immunohistochemistry (IHC) and ISH have not been sufficiently standardized for routine use in the identification of fungi because of the wide technical variation of their methods [3,14].

Mucormycosis is a fatal IFI, especially after hematopoietic stem cell transplantation (HSCT). As we recently encountered an autopsy case with disseminated mucormycosis that was not confirmed before death, we assessed the usefulness of several supplemental diagnostic procedures, specifically PCR, IHC, and ISH, in this case. The difference in the detection of fungal infection was compared among the techniques. We would like to consider an efficient approach for diagnosing IFI in uncultured autopsy FFPE samples based on this result.

## 2. Case Description

A 54-year-old man underwent chemotherapy with a diagnosis of T lymphoblastic lymphoma more than two years before this admission and then underwent human leukocyte antigen (HLA)-matched allogeneic HSCT at another hospital approximately one year later. After HSCT, the underlying disease was controlled. However, in addition to poorly controlled graft-versus-host disease (GVHD), he experienced repeated bacterial pneumonia/bloodstream infection, sinusitis, and cytomegalovirus pneumonia.

The patient was transferred to our hospital to treat chronic GVHD (day 0). On day +2, he developed pneumonia and bacteremia caused by *Pseudomonas aeruginosa*, and he was treated with broad-spectrum antibacterial agents. At the same time, his GVHD worsened, and the use of immunosuppressive agents (calcineurin inhibitors) increased, including increased doses of steroids. After that, he continued to experience repeated episodes of bacteremia caused by indigenous skin bacteria, and broad-spectrum antibacterial agents were used each time. The blood test on day +35 confirmed positivity for the Aspergillus galactomannan antigen (+, 0.5), whereas testing for β-d glucan was negative. Breakthrough infection was a concern because he received voriconazole as a prophylactic prior to hospitalization, and liposomal amphotericin B was considered for antifungal use. However, there was a risk of worsening kidney function, and micafungin was administered from day +37 until two days before his death. In addition, he underwent endoscopic sinus surgery for chronic sinusitis at the otolaryngology department three weeks before his death (day +37). However, no bacterial mass or pus was collected, and there were no findings of fungal infection. Two weeks before his death, the patient was transferred to the intensive care unit because his level of consciousness declined, and dyspnea appeared (day +44). At this point, a cavity shadow in the left upper lung field was visible on the chest X-ray (Figure 1A). Despite intensive treatment, multiple organ failure and disseminated intravascular coagulation progressed, and his respiratory condition worsened, leading to death attributable to respiratory failure (day + 58). Throughout this hospitalization, no fungi were detected from any cultures.

An autopsy excluding the brain was performed. The section of the myocardium revealed several foci of hemorrhagic necrosis measuring up to 15 mm in size at the posterior–lateral wall of the left ventricle. Furthermore, a verruca was found on the mitral valve. In addition to congestive edema in the lungs, a section of the upper lobe of the left lung revealed a hemorrhagic infarct measuring 80 × 60 mm^2^ in size. There were foci of necrosis in the spleen and three hemorrhagic ulcers of less than 100 mm in diameter on the large intestinal mucosa (Figure 1B–F). Histological examination revealed filamentous fungi in all of the aforementioned foci. The case was diagnosed as “mucormycosis” based on the typical hyphae morphology” (Figure 2 and Appendix A).

## 3. Materials and Methods

### 3.1. Histopathology

All sections prepared from FFPE blocks were mounted on slide glasses, deparaffinized, stained with hematoxylin and eosin, periodic acid-Schiff (PAS), and Grocott’s methenamine silver (GMS). The histopathological diagnosis of IFI was made by three expert pathologists using light microscopy. The morphological features of fungi referred to “Histopathologic diagnosis of fungal infections in the 21st century” [1]. FFPE blocks of each organ (heart, lungs, liver, spleen, kidneys, colon) containing the most fungal elements were selected for this identification. To present the number of hyphae in the tissue sections, the ratio of the area of each section, in which the total tissue area was the denominator, and the number of fungal components present was the numerator, was described as a percentage. Because the density of invading hyphae differed in each organ, we classified the density of fungal components to three degrees as low, medium, or high. All samples from each organ were verified by PCR, IHC, and ISH. To assess the reactivity for the IHC and ISH methods, we prepared FFPE sections of collected hyphae of *Aspergillus fumigatus*, *Cunninghamella elegans*, and *Rhizopus oryzae* from their cultures.

### 3.2. DNA Extraction, PCR, and Sequencing

Six 10-μm-thick sections taken from each FFPE block were used. According to the manufacturer’s instructions, DNA extraction was performed using a QIAamp DNA FFPE Tissue Kit (Qiagen, Duesseldorf, Germany). Four panfungal PCR primers targeting the ITS region (ITS1–ITS2, ITS3–ITS4, ITS1–ITS4) [15], two panfungal PCR primers targeting the D1/D2 region of fungal rDNA (NL1–NL4) [16], and three primers targeting 18S rRNA of mucormycosis-causing fungi (ZM1–ZM2, ZM1–ZM3) [17], all of which were described previously, were tested in the present study. The human housekeeping gene (GAPDH) was used as a quality control of the nucleic acid extraction. The nucleotide sequences of primers targeting the ITS region were as follows: ITS1, 5′-TCC GTA GGT GAA CCT GCG G-3′; ITS2, 5′-GCT GCG TTC TTC ATC GAT GC-3′; ITS3, 5′-GCA TCG ATG AAG AAC GCA GC-3′; and ITS4, 5′-TCC TCC GCT TAT TGA TAT GC-3′. The nucleotide sequence of primers targeting the D1/D2 region were as follows: NL1, 5′-GCA TAT CAA TAA GCG GAG GAA AA-3′; and NL4, 5′-GGT CCG TGT TTC AAG ACG G-3′. The nucleotide sequence of primers targeting the 18S rRNA of fungi-causing mucormycosis were as follows: ZM1, 5′-ATT ACC ATG AGC AAA TCA GA-3′; ZM2, 5′- TCC GTC AAT TCC TTT AAG TTT C-3′; and ZM3, 5′-CAA TCC AAG AAT TTC ACC TCT AG-3′. The nucleotide sequence targeting GAPDH were as follows: GAPDH-F, 5′-ACC ATG GAG AAG GCT GGG G-3′; and GAPDH-R, 5′-CAA AGT TGT CAT GGA TGA CC-3′. All primers were synthesized by Eurofins Genomics (Tokyo, Japan).

The methods of PCR and sequencing used in this study were previously described [18]. Briefly, PCR was performed using a T100 thermal cycler (Bio-Rad Laboratories, CA, USA). Each PCR mixture contained 5 μL of extracted DNA, 1 μL of 1.25 U/μL MightyAmp DNA Polymerase (Takara Bio Inc., Shiga, Japan), 0.5 μL of each primer (100 μM), 25 μL of 2× MightyAmp Buffer Ver.2 (Takara Bio Inc., Shiga, Japan), and distilled water up to 50 μL. The PCR mixture was first denatured at 98 °C for 2 min, followed by 40 cycles of denaturation at 98 °C for 10 s, annealing at 55 °C for 15 s, and extension at 68 °C for 30 s, and then final extension at 68 °C for 3 min. The PCR products were electrophoresed on a 1.5% agarose gel and visualized under a Blue/Green LED light. Amplicons of positive samples were purified using a QIAquick PCR purification kit (Qiagen) and sequenced by Eurofins Genomics. The sequence result was compared with those in the MycoBank or GenBank database. A similarity rate > 99% for amplicons was required to confirm the identification of fungi.

### 3.3. IHC

We performed IHC using commercially available antibodies. The primary antibodies were an anti-*Aspergillus* rabbit polyclonal antibody (ab20419, Abcam, Cambridge, UK) and an anti-*R. arrhizus* mouse monoclonal antibody (clone WSSA-RA-1, GeneTex, Los Angeles, CA, USA).

Staining with the anti-*Aspergillus* antibody was performed using Ventana Ultra (Roche Diagnostics, Basel, Switzerland), an autostainer for IHC. However, staining with the anti-*R. arrhizus* antibody was performed manually using an IHC Detection Kit Micropolymer (ab236466, Abcam, Cambridge, UK) because the autostainer was difficult to adapt for this antibody. 

Briefly, 4-μm-thick FFPE sections were prepared from each FFPE block as described for PCR. For the manual protocol, after deparaffinization and rehydration were performed using standard protocols, a hydrogen peroxidase block (ab236466, Abcam, Cambridge, UK) was performed. After, a protein block (ab236466, Abcam, Cambridge, UK) was performed for 10 min at room temperature. The sections were incubated with primary anti-*R. arrhizus* antibody (1:100) at room temperature for 1 h. The procedure from secondary antibody incubation to visualization followed the manufacturer’s instructions (ab236466, Abcam, Cambridge, UK). IHC using an autostainer was performed using the manufacturer’s recommended instructions. Briefly, slides were heat-treated for 1 h with Cell Conditioner 1 buffer (Roche Diagnostics, Basel, Switzerland) for antigen retrieval. The incubation time of the primary anti-*Aspergillus* antibody (1:300) was 32 min in the autostainer protocol. An optiview universal DAB kit (Roche Diagnostics, Basel, Switzerland) was used for visualization.

### 3.4. ISH

#### 3.4.1. ISH Probes Used in This Study

Three types of probes, namely peptide nucleic acid (PNA), double-stranded DNA (dsDNA), and oligonucleotide DNA probes were used in this study. First, three different panfungal probes were examined in all organs to compare the retention and hybridizability of the target nucleic acid for each probe. Meanwhile, the remaining six probes were examined in the heart sections containing the highest fungal load, whereas only positive ones were examined in the other organs. 

1. PNA probe; panfungal

The target gene of 28S ribosomal RNA genes of the panfungal PNA probe was 5′-TAC TTG TGC GCT ATC GGT-3′, which was the same as that used in previous reports [19]. FITC was conjugated to the 5′-terminus of the PNA probe. The panfungal PNA probe was synthesized by Fasmac Co., Ltd. (Kanagawa, Japan). 

2. dsDNA probe; *Aspergillus*, panfungal, mucormycosis (18S rRNA), mucormycosis (28S rRNA)

All probes were digoxigenin (DIG)-labeled when PCR was performed. DIG labeling and synthesis of each probe were performed by first mixing 50 μL of 2× KAPA2G Fast HotStart ReadyMix with dye (KAPA Biosystems, Inc., MA. US), 5 μM DIG-11-dUTP (Roche Diagnostics, Basel, Switzerland), 100 pmol of each primer, 100 ng of template extracted DNA, and distilled water to a volume of 100 μL. The thermocycling conditions were as follows: initial denaturation at 95 °C for 3 min, 30 cycles of denaturation at 95 °C for 15 s, annealing at 60 °C for 15 s, and extension at 72 °C for 15 s, and final extension at 72 °C for 10 min. The PCR products were purified using a QIA quick PCR purification kit (Qiagen).

The 583-bp dsDNA probe targeting *Aspergillus* spp. used in this study was previously reported by Hanazawa et al. [20] with slight modification. The primers used were as follows: ALP-ns, 5′-GCC TAT CCG TGT ACT TGA TG-3′; and ALP-nr, 5′-GTT GAT CGT GCT GAA CCT T-3′, which target the alkaline protease of *A. fumigatus*. *A. fumigatus* (NBRC 6344) DNA was used as a template for the dsDNA probe.

The sequence used to synthesize the panfungal dsDNA probe (673 bp) referred to that previously reported by Makimura et al. [21]. The primers used were as follows: B2F, 5′-ACT TTC GAT CGT AGG ATA G-3′; and B4R, 5′-TGA TCG TCT TCG ATC CCC TA-3′. *C. albicans* (ATCC10231) DNA was used as a template for the dsDNA probe. The technique used to synthesize followed the aforementioned procedure.

The sequence used to synthesize the dsDNA probe targeting 18S rRNA of mucormycosis-causing fungi (156 bp) referenced a previous report by Fujiwara et al. with a slight modification of the primers used [22], which were as follows: Muc11F, 5′-AGT TAA AAC GTC CGT AGT CAA A-3′; and Muc7R, 5′-AAC ACT CTG ATT TGC TCA TGG T-3′. The DNA of *R. arrhizus* (IFM40515) was used as a template for the dsDNA probe. The technique used for synthesis followed the aforementioned protocol.

The sequence used to synthesize the dsDNA probe targeting 28S rRNA of mucormycosis (247 bp) was originally designed to detect pan-mucormycosis rather than being specific for *Cunninghamella* spp. This probe was based on a multiple alignment comparison of the 28S rRNA gene sequences of fungi that may cause mucormycosis to identify regions with high homology in Mucorales. The primers used were as follows: muc-f, 5′-TGT GAA ATT GTT AAA AGG GAA C-3′; and muc-r, 5′-ACC GTA GTA CCT CAG AAA ACC T-3′. The DNA of *R. arrhizus* (IFM40515) was used as a template for the dsDNA probe. The technique used for synthesis followed the aforementioned protocol.

3. Oligonucleotide DNA probe 

According to a previous report by Hayden et al., the target gene of 18S rRNA was selected [23]. The nucleotide sequences of probes were as follows: *Mucor* spp., *Rhizomucor* spp., *Rhizopus* spp., and *Saksenaea* spp., 5′-TCA ATG AAG ACC AGG CCA C-3′; *Absida* spp., 5′-ACC TGA CCA AAG GTC AAG GC -3′; *Cunninghamella* spp., 5′-TGG CTA GAC CGA AAT CTA GAA AC -3′. DIG was conjugated to the 3′-terminus of each oligonucleotide probe. All four oligonucleotide DNA probes were synthesized by Eurofins Genomics.

#### 3.4.2. ISH Using the PNA Probe

ISH was performed using the method previously described by Shinozaki et al. [19] with minor modifications. Briefly, 4-μm-thick sections of FFPE tissue from each organ were prepared for ISH. The sections were deparaffinized and rehydrated using standard procedures. The sections were incubated with antigen retrieval solution (pH 9.0, Nichirei Biosciences, Inc., Tokyo, Japan) in a water bath at 98 °C for 20 min to expose target nucleic acids and then incubated with 10 μg/mL proteinase K (FUJIFILM Wako Pure Chemical Corporation, Osaka, Japan) at 37 °C for 10 min. Hybridization was performed using 1 μg/mL PNA probe dissolved in hybridization buffer (Nippon Gene Co., Ltd., Tokyo, Japan) for 90 min at 56 °C. After repeated washing at 56 °C with 2× standard saline citrate (SSC) followed by 0.2× SSC, the sections were incubated with anti-FITC antibody (Roche Diagnostics, Basel, Switzerland) and horseradish peroxidase-labeled polymer solution (Nichirei Biosciences, Inc., Tokyo, Japan). Finally, the peroxidase site was visualized using 3,3′-diaminobenzidine tetrahydrochloride (DAB; Dojindo Research Institute, Kumamoto, Japan) in the presence of hydrogen peroxide, nickel ions, and cobalt ions.

#### 3.4.3. ISH Using dsDNA and Oligonucleotide DNA Probes

The ISH technique applied in this study followed the method previously described by Hanazawa et al. with minor modifications [20]. First, 4-μm-thick tissues were prepared, deparaffinized, and rehydrated using standard procedures. After washing with distilled water, tissue sections were treated with 0.2 N HCl for 10 min at room temperature, washed with distilled water, and digested with 2 μg/mL proteinase K in phosphate-buffered saline (PBS) for 5 min at 37 °C. To inactivate proteinase K, tissue sections were immersed in PBS containing 2 mg/mL glycine for 10 min. For the next step, the tissue sections were covered with 100 μL of hybridization buffer (1× Denhardt’s solution [Sigma-Aldrich Japan, Tokyo, Japan], 10 mM Tris-HCl, pH 7.6, 50% formamide, 10% dextran sulfate, 600 mM NaCl, 0.25% SDS, 1 mM EDTA, pH 8.0), heated on a 94 °C hotplate for 10 min, immediately cooled on ice, and prehybridized at 50 °C for 1 h. DIG-labeled probes were heated in boiling water for 10 min and immediately cooled in ice water. Hybridization buffer (100 μL) containing 2 μg/mL of each probe was applied to tissue sections and covered with Parafilm (Bemis Company, Neenah, Wisconsin, USA). Hybridization was performed overnight at 50 °C. After hybridization, tissue sections were washed twice with 2× SSC for 15 min at 50 °C, followed by 0.2× SSC using the same conditions to remove unhybridized and mismatched probes. Then, sections were blocked with 10% blocking reagent (Roche Diagnostics, Basel, Switzerland) in buffer 1 (100 mM Tris-HCl, 150 mM NaCl, pH 7.5) for 1 h. The hybridization products were detected using an anti-DIG-alkaline phosphatase conjugate antibody (Roche Diagnostics, Basel, Switzerland). Anti-DIG agent (diluted 500-fold in buffer 1) was applied to tissue sections and allowed to react for 30 min at room temperature, followed by three washes for 15 min each with buffer 1 containing 0.2% Tween 20 to remove the excess anti-DIG agent. Finally, to visualize the signals, an alkaline phosphatase conjugate substrate kit (170-6432, Bio-Rad, Hercules, California, USA) was used as the substrate for alkaline phosphatase overnight at 37 °C in a light-shielded box. After colorization, sections were washed with distilled water to stop the reaction. The stained hybridization slides were dehydrated, cleared, and mounted with cover glasses for microscopic observation.

## 4. Results

The results of this study are summarized in Table 1. Fungal components were most abundant in sections of the heart, followed by the lungs. Relatively, there were fewer fungal elements in the kidney and colon (Appendix A). Foci in the spleen and liver were extensively necrotic among the six organs examined in this study, and the spleen was judged to be negative for infection by PCR. Using the panfungal primer (NL1–NL4), Polyporaceae spp. were detected only in the heart, probably as a contaminant, whereas PCR using ZM1–ZM3, which has a short (approximately 170 bp) target gene sequence, detected the same gene homologous to *Cunninghamella elegans* (100% similarity to various strains including GenBank ID: AH009069, AF113422.1) in organs excluding the spleen.

All organs examined by IHC were negative for antibodies against *A. fumigatus* but positive for antibodies against *R. arrhizus* (Figure 3). The anti-*Aspergillus* antibody showed apparent reactivity to the cultured *A.fumigatus* compared to others. On the contrary, the results of the anti-*Rhizopus arrhizus* antibody in IHC for cultured fungi were all reactive to varying degrees (Appendix A).

ISH revealed positive staining in all organs for two of the three panfungal probes (PNA and dsDNA probes) examined in this study, whereas positive staining was not detected in any organs using the oligonucleotide DNA probe (Figure 4). Of the remaining six ISH probes examined in this study, two-dsDNA probes targeting *Mucor* spp. and an oligonucleotide DNA probe targeting *Cunninghamella* spp. showed positive signals on the hyphae. Conversely, the dsDNA probe targeting aspergillosis-causing fungi and the oligonucleotide DNA probe-target mucormycosis-causing species other than *Cunninghamella* spp. were negative (Figure 4). Thus, excluding the oligonucleotide DNA probe targeting *Cunninghamella* spp., the ISH method did not display significant differences in detection among the examined organs. However, no signals were confirmed in hyphae in foci with conspicuous necrosis in the background (spleen and liver), but cases that showed positive signals were present even in a part of the hyphae were deemed positive and annotated as “focal.” The results of ISH probes applied to cultured fungi are shown in Appendix A. The 18S rRNA panfungal probe showed a better reaction to cultured fungi than others. Except for *A. fumigatus* culture specimen, the reactivity of the species-specific probe was generally good. The species-specific oligoneucleotide probes seemed reasonable specificity, although the response was weak.

## 5. Discussion

In the present autopsy case, the patient was diagnosed with disseminated mucormycosis after death. Because the fungal infection was not proven by endoscopic examination of the paranasal sinuses before death, the fact that large foci developed in both respiratory and gastrointestinal tracts suggested that these routes could be the portals of entry in this case.

As fungal infection was found in systemic organs, including endocarditis with vegetation, numerous hyphae might be circulated via the bloodstream, but the blood culture was negative. In post-HSCT patients such as the current case, we sometimes observe breakthrough infection (mucormycosis) during treatment with antifungal drugs such as voriconazole and micafungin, even if blood cultures are negative. Infectious diseases remain the leading cause of death in post-HSCT patients, and IFI has been known as a common disease initially confirmed by the autopsy [11].

Previous studies conducting epidemiological retrospective observation using autopsies provided clinically important data of IFI including mucormycosis [12,13,24]. In those epidemiological studies, the diagnose of IFI was usually made by culture or histologically. Although histopathological diagnosis using FFPE specimens is widely used for the diagnosis of IFI because of its rapidity, simplicity, and versatility, there are some concerns regarding its use in the histopathological diagnosis of fungal infection. Specifically, because histopathological diagnosis is based on the shape of fungi in tissue, identifying some species might be difficult only by assessment of the shape, and the accuracy of diagnosis could largely depend on the pathologist’s skill [1,25]. Conversely, the advantage of molecular biological analysis techniques such as PCR is that the differences among facilities can be minimized using the same primers and other materials used for analysis, thus creating objectivity in diagnosis [3,26].

It is expected that additional studies using PCR for identification should be conducted in the future because the latest EORTC/MSGERC guidelines emphasized the usefulness of PCR for identifying causative fungal species using FFPE samples [3]. However, we believe the approach for identifying IFI in autopsy FFPE samples should be different from those using FFPE samples from biopsy and surgery because of its aforementioned rigidity. Few studies have focused on the reliability of PCR and ISH using FFPE samples obtained by autopsy [7,18,27]. Therefore, we decided to consider an efficient approach for diagnosing IFI in autopsy FFPE samples in this study.

In addition to the primer sets targeting the fungus ITS and D1/D2 regions recommended in the guidelines, the present study also employed the primers targeting the 18S rRNA of zygomycetes for PCR analysis. In identification using panfungal primers targeting the D1/D2 region (NL1–NL4), the detected fungi appeared to be contaminants because they were detected only in a single organ. By contrast, PCR using the mucormycosis-specific primer set produced positive results in five of six organs, and sequences of *Cunninghamella elegans*, which were also consistent with the histopathological diagnosis, were detected in all five organs. The PCR results from the spleen were negative, but it is believed that the strong necrosis of the background tissue may have influenced the finding.

As previously summarized in detail by Dannaoui et al., it has been confirmed that the detection of fungus observed sections from FFPE samples by PCR increases with the number of fungi present [26]. However, even if PCR is using autopsy FFPE samples with many fungal elements, we found that the result of PCR using panfungal primers may not always be positive, as demonstrated in the present study. As causes, it has been stated that decay of the specimen itself and an excessive period of formalin fixation can increase gene fragmentation in FFPE tissue and that the administration of antifungal drugs may affect the extraction yield of fungal DNA [28,29]. When handling autopsy FFPE samples, for the aforementioned reasons, PCR enzymes with strong amplification capacity and shorter PCR product are necessary to improve sensitivity. Furthermore, because autopsy FFPE samples are not aseptic, contamination is apt to occur when using the panfungal primers recommended in the guidelines [3,30].

Based on the aforementioned results, we believe that the fungus identified by PCR and sequencing should be consistent with the histopathological diagnosis, and the same gene sequence should be detected in multiple sections when applying PCR to analyze autopsy FFPE samples when contamination cannot be avoided. In addition to panfungal primers, the use of species-specific primer sets with shorter amplicons (<200 bp) might be considered to improve detection.

Concerning IHC, we used commercially available antibodies that have been reported in several studies, including a veterinarian autopsy case report of *Cunninghamella bertholletiae* in which the same anti-*R. arrhizus* antibody produced positive results [31,32,33]. These studies concluded that IHC using commercially available antibodies might be helpful for differentiating aspergillosis from mucormycosis. From our IHC experience in this study, anti-*R. arrhizus* antibody produced positive results in all organs, whereas negative results were obtained using the anti-*Aspergillus* antibody. Although, the results for anti-*R. arrhizus* antibody in IHC for cultured fungi were all positive to varying degrees (Appendix A). It can be inferred that IHC is not inadequate for the differential detection of *Cunninghamella* spp. and *Aspergillus* spp. in FFPE tissues. Because the antigens of many commercially available antibodies have not been clarified, it is practically impossible to examine the cross-reactivity of each antibody with several fungi in the real world. However, the antibodies may be used as an adjunct in the histopathological diagnosis of aspergillosis and mucormycosis to determine the genus, and the staining of background organs is easily maintained even in areas of necrosis. In addition, because automated immunization systems such as those used in this study are widely available, if highly sensitive and specific antibodies for fungi are produced, IHC will be worthy of consideration in the future.

Although several studies used ISH to identify fungi in FFPE materials, there is little evidence to date regarding the use of ISH for detecting mucormycosis [34]. The oligonucleotide DNA panfungal probe produced negative results, whereas positive results were obtained using the dsDNA and PNA panfungal probes. However, the oligonucleotide DNA panfungal probe, which produced negative results in the case study, produced positive staining, albeit weakly, in the strain study (Appendix A). Oligonucleotide DNA is weaker than dsDNA and PNA in terms of binding to nucleic acids, and thus, the difference in the results may be attributable to differences in target gene binding depending on the type of probe. From this result, compared with other probes, oligonucleotide DNA probes bind DNA more weakly in autopsy FFPE tissues. It is reasonable to suppose that conventional special stains such as PAS and GMS are better for fungal detection in terms of both sensitivity and specificity. Although there are few opportunities for the ISH method using pan-fungal probes for fungal screening, we believe that ISH using panfungal probes is meaningful as a positive control when using this method.

Only a limited number of studies of ISH targeting *Cunninghamella* spp. are available. In one study, Hayden’s group used the same probe applied in our investigation [23]. Their paper stated that it is difficult to identify mucormycosis-causing fungi using ISH, unlike other *Candida* and *Aspergillus* species. It is challenging to design pan-Mucorales probes to target all species causing mucormycosis with a wide range of genetic polymorphisms.

The two-dsDNA probes targeting mucormycosis-causing fungi examined in this study were previously designed to target the more common species such as *Rhizopus* and *Mucor* spp. Although the probes were not mainly designed to target *Cunninghamella* spp., they produced positive results. However, these two-dsDNA probes targeting mucormycosis showed a positive reaction against *A. fumigatus* culture (Appendix A). On the contrary, the dsDNA probes targeting *Aspergillus fumigatus* alkaline proteinase gene showed good reactions in both the case study and cultures. Therefore, in cases of mucormycosis, for which differentiation from aspergillosis tends to be difficult in actual clinical practice, we believe that one of the auxiliary diagnoses is to confirm that ISH for aspergillosis is negative using probes for both mucormycosis-causing fungi and other species such as *Aspergillosis* spp. Oligonucleotide probes designed to differentiate between the causative species of several forms of mucormycosis produced valid findings for cultures (Appendix A). In addition, in the examination of oligonucleotide DNA probes used to differentiate the causative organism of mucormycosis, only the probe for *Cunninghamella* spp. was found to produce positive results in some organs in this case, which was believed to be associated with *Cunninghamella elegans*. The reason why the oligonucleotide probe generated positive results in only some organs might be the low binding power of the probe itself, as inferred from the investigation of the probe for panfungal detection. In fact, Hirayama et al. reported that positive findings were observed in an autopsy mucormycosis case analyzed by designing an oligonucleotide RNA probe using the same gene sequence as the oligonucleotide DNA probe examined in the current study [35]. However, because this report did not provide evidence of fungal identification by other methods such as culture or PCR, it is considered insufficient to examine the probe based on this result alone, but adapting the same sequence to different types of probes may be helpful for improving the examination.

A limitation of this study was that only a single case of mucormycosis was investigated, and the results alone are insufficient to standardize the analysis of autopsy FFPE samples. Based on this result, we believe that it is necessary to accumulate more cases in the future to standardize the analysis of autopsy FFPE samples.

In conclusion, although it is difficult to use a single ISH probe for species identification, the combination of multiple ISH probes may lead to identifying the genus and species of the causative organism. In addition, because of the wide variety of species associated with mucormycosis, we believe it may be realistic to determine only the genus using ISH in autopsy FFPE samples. It might be worth noting that fungi were detected by IHC and ISH methods in the section of the spleen that was negative by PCR. This fact points us to emphasize the usefulness of IHC and ISH in tissues with conspicuous necrosis. Table 2 provides a list of current approaches for identifying fungi in FFPE samples. Each method has both advantages and disadvantages. We think some of the techniques can compensate for each other and improve diagnosis. The latest report from Liu et al. has applied real-time PCR, ISH, and IHC to diagnose mucormycosis from FFPE clinical samples [36]. They conclude that the combined use of these techniques enables accurate diagnosis of mucormycosis in FFPE specimens. At present, our findings make it reasonable to recommend PCR with sequencing to identify the causative fungal species for postmortem histologically proven cases of IFI cases. However, using autopsy FFPE samples, PCR sometimes produces negative results. In such cases, the ISH and IHC methods discussed in this report may be worth considering for detecting the broad classification of the fungus, such as the genus- or species-level classification.

## Figures and Tables

**Figure 1 jof-08-00337-f001:**
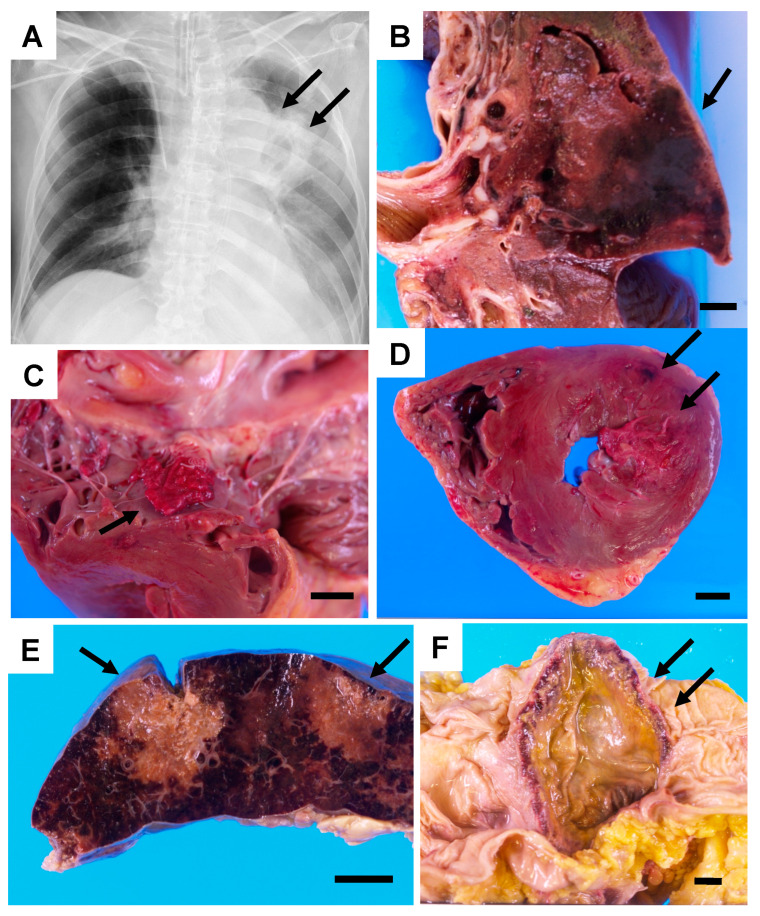
Chest X-ray and autopsy macroscopic findings of the autopsy case: (**A**) Chest X-ray on day 53. An infiltrative shadow with cavity formation is present in the left upper lung field (arrows). (**B**) The cut surface of the formalin-fixed upper left lung. Hemorrhagic infarction measuring 80 × 60 mm^2^ in size (arrow). (**C**) Vegetation of the mitral valve (arrow). (**D**) The horizontal cut surface of the heart. Ischemic change with hemorrhage in the posterior–lateral wall of the left ventricle (arrows). (**E**) Cut surface of the formalin-fixed spleen. Necrotic foci are observed (arrows). (**F**) Formalin-fixed colon. Tumor-like ulceration up to 100 mm in size is present in the transverse colon (arrows). Each scale bar in the figure is 10 mm.

**Figure 2 jof-08-00337-f002:**
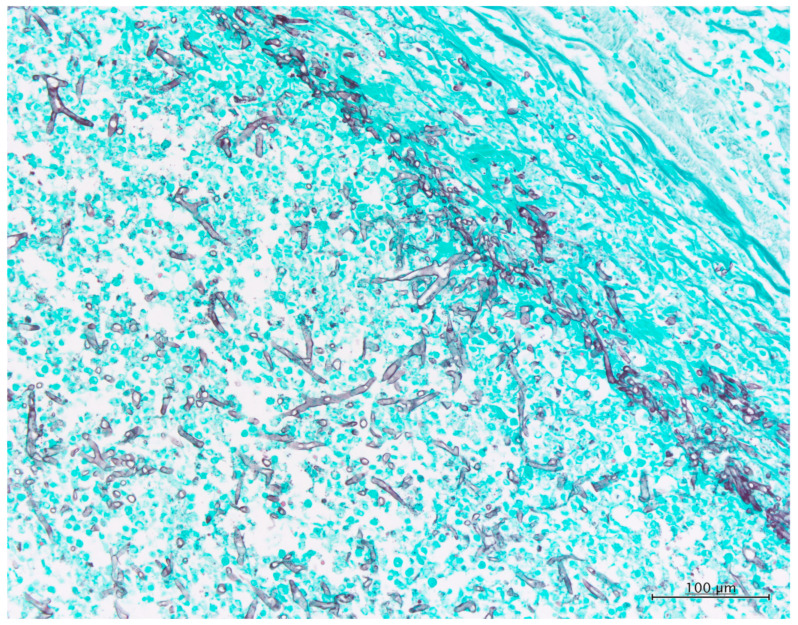
Histopathology findings of the autopsy case of the lung. Grocott’s stain (×200). Many hyphae in the lumen of the pulmonary artery have a thin and folded wall and show bifurcation with irregular angle, but rectangular bifurcation is occasionally noted. Septae are rarely found.

**Figure 3 jof-08-00337-f003:**
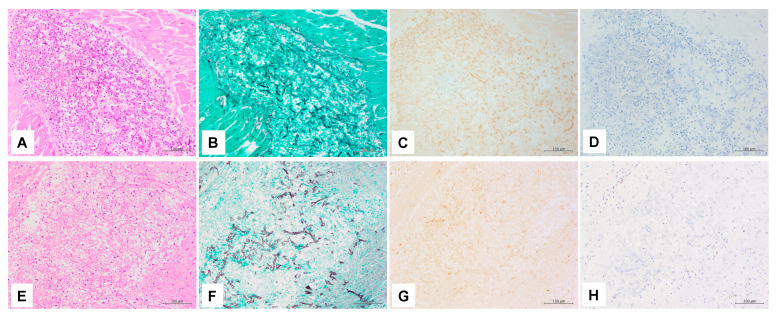
Immunohistochemistry findings: Pictures (**A**–**D**) were taken from the same heart area. Pictures (**E**–**H**) were taken from the same spleen area. Hematoxylin eosin (HE) staining (**A**,**E**), Grocott’s methenamine silver (GMS) staining (**B**,**F**), (**C**) Positive reaction for anti-*Rhizopus arrhizus* antibody. Brown staining of hyphae can be seen in the heart tissue. (**D**) Negative reaction for anti-*Aspergillus* antibody. (**G**) Positive reaction for anti-*Rhizopus arrhizus* antibody. (**H**) Negative reaction for anti-*Aspergillus* antibody. Magnification of all pictures are (200×).

**Figure 4 jof-08-00337-f004:**
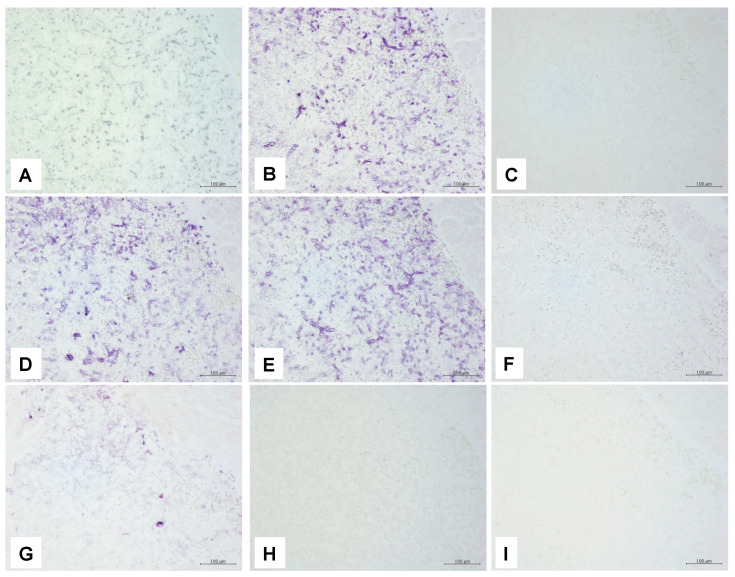
In situ hybridization findings. All the pictures in this figure are taken from the same heart area as IHC presented in Figure 3A–D. (**A**) Positive reaction for PNA panfungal probe. Dark brown pigmentation attributable to diaminobenzidine staining is present in line with hyphae. (**B**) Positive reaction for 18S rRNA panfungal probe. Purple pigmentation attributable to alkaline phosphatase staining is present in line with hyphae. (**C**) No reaction is observed for the oligonucleotide DNA panfungal probe. (**D**) Positive reaction for the 18S r RNA mucormycosis probe. Purple pigmentation attributable to alkaline phosphatase staining is present in line with hyphae. (**E**) Positive reaction for the 28S r RNA mucormycosis probe. (**F**) Negative reaction for the probe targeting alkaline proteinase of *Aspergillus fumigatus*. (**G**) Positive reaction for the oligonucleotide DNA probe targeting *Cunninghamella* spp. The positive signal is weaker than those of two mucormycosis probes (**A**,**B**). (**H**) Negative reaction for the oligonucleotide DNA probe targeting *Absida* spp. (**I**) Negative reaction for the oligonucleotide DNA probe targeting *Mucor* spp., *Rhizomucor* spp., *Rhizopus* spp., and *Saksenaea* spp. Magnification of all pictures are (200×).

**Table 1 jof-08-00337-t001:** The histopathological evaluation of each tissue organ and PCR, IHC, and ISH results.

Tissue Organ			Heart	Lung	Liver	Spleen	Kidney	Colon
Histopathology	Area of Fungal element/Total area of FFPE		70%	60%	50%	80%	50%	40%
	Density of fungal components		High	Medium	Low	Medium	Low	Low
	Necrosis background		−	−	+	+	−	−
PCR	Panfungal	ITS1-ITS2	−	−	−	−	−	−
		ITS3-ITS4	−	−	−	−	−	−
		ITS1-ITS4	−	−	−	−	−	−
		NL1-NL4	+	−	−	−	−	−
	Mucorales	ZM1-ZM2	−	−	−	−	−	−
		ZM1-ZM3	+	+	+	−	+	+
IHC	Anti-Aspergillus antibody	ab20419	−	−	−	−	−	−
	Anti*-Rhizopus arrhizus* antibody	WSSA-RA-1	+	+	+	+	+	+
ISH	Panfungal	PNA probe	+	+	+(focal)	−	+(focal)	+(focal)
		18S rRNA probe	+	+	+	+(focal)	+	+
		Oligo nucleotide DNA probe	−	−	−	−	−	−
	Mucormycosis	28S rRNA probe	+	+	+	+	+	+
		18S rRNA probe	+	+	+	+	+	+
	Aspergillosis	*A. fumigatus* alkaline proteinase gene	-	NT	NT	NT	NT	NT
	*Mucor, Rhizomucor, Rhizopus, Saksenaea*	Oligonucleotide DNA probe	−	NT	NT	NT	NT	NT
	*Absida*		−	NT	NT	NT	NT	NT
	*Cunninghamella*		+	−	+	−	+	+

Formalin-fixed paraffin-embedded: FFPE, polymerase chain reaction: PCR, immunohistochemistry: IHC, in situ hybridization: ISH, NT: not tested, +: positive, −: negative.

**Table 2 jof-08-00337-t002:** The available approaches for identifying fungal species in FFPE.

Method	Accessibility	Standardization	Time for Detection	Cost	Automation Technique	Advantage	Disadvantage
Histopathology	Easy	Yes	Minutes	Low	Yes	Low cost, Fast, High sensitivity.	Difficult for identifying up to species level.
IHC	Easy-Limited	No	Hours	Intermediate	Yes	It can be fully automated. Good sensitivity.	Commercially available antibody is few. Specificity depends on the quality of the antibody.
ISH	Limited	No	Hours-days	Intermediate	Partial or no	Good sensitivity with good specify.	Takes time. Specific probes needed for detecting different species.
Broad range PCR with sequencing	Easy-Limited	Yes	Hours-day	Intermediate	Partial or no	Can be identify up to the species level.	Contamination of ubiquitous fungi can happen.
Real-time PCR	Limited	Yes	Hours	High	Partial	The result can be shown by real-time with quantification.	Detectible species depend on the primer use.
NGS	Limited	No	Hours	High	Partial	Can potentially detect any fungal pathogen.	Limited evidence. High cost and special equipment should be needed.

Formalin-fixed paraffin-embedded: FFPE, immunohistochemistry: IHC, in situ hybridization: ISH, next-generation sequencers: NGS.

## Data Availability

The publicly available data are presented with the manuscript text. Further information is available on request from the corresponding author.

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
