# Peer review of "Comparison Approach for Identifying Missed Invasive Fungal Infections in Formalin-Fixed, Paraffin-Embedded Autopsy Specimens"

_jof, 2022, doi:10.3390/jof8040337_

Round 1
Reviewer 1 Report
Please describe HLA HSCT
Fig 1 scale bars are missing, the regions of interest are not marked
Fig 2 corresponding HE, PAS is missing, which tissue is shown, e.g., lung tissue or heart? Please use every time the same magnification as in Figure 4.
Figure 3 would better present the ROI with a higher magnification (like Figure 4). Also depict the corresponding HE, PAS, and Grocott’s.
Figure 4 and 5 may also depict the corresponding HE, PAS, Grocott’s and IHC.
Maybe it would be clearer if you put pictures 3, 4 and 5 together? And also, add the corresponding HE, PAS, and Grocott’s.
S 11 line 371 please fix the sentence fragment: Invasive mucormycosis is one of the
Author Response
Thank you for your comment and their valuable suggestions. We updated the manuscript.
Comment from Reviewer 1:
- Please describe HLA HSCT
Our response: In accordance with the reviewer's comment, we spell out HLA as Human Leukocyte Antigen (HLA) in Line 84.
- Fig 1 scale bars are missing, the regions of interest are not marked
Our response: As per your suggestion, we put scale bars and arrows for explanation to Fig 1.
- Fig 2 corresponding HE, PAS is missing, which tissue is shown, e.g., lung tissue or heart? Please use every time the same magnification as in Figure 4.
Our response: The tissue shown in Fig 2 is the lung. We changed the magnification of Fig 2 same as Fig 4. All the figure in the main text is the same magnification. In this Fig 2, we would like to clearly show the filament shape of fungi. Therefore, other staining such as HE and PAS are together presented as a summary of histopathology in Figure S7.
- Figure 3 would better present the ROI with a higher magnification (like Figure 4). Also depict the corresponding HE, PAS, and Grocott’s.
Our response: We followed your suggestion; we decided to present HE and Grocott’s together with ROI with the same magnification as ISH. PAS staining is shown in Figure S7.
- Figure 4 and 5 may also depict the corresponding HE, PAS, Grocott’s, and IHC.
Our response: The area presented in Figures 3 and 4 is the same area of the heart. The summary of histopathology, which contains HE, PAS, Grocott’s, is shown in Figure S7.
- Maybe it would be clearer if you put pictures 3, 4, and 5 together? And also, add the corresponding HE, PAS, and Grocott’s.
Our response: We followed your suggestion; we decided to put Fig 4 and Fig 5 together. The summary of histopathology, which contains HE, PAS, Grocott’s, is presented in Figure S7.
- S 11 line 371, please fix the sentence fragment: Invasive mucormycosis is one of the
Our response: Thank you for your suggestion. We delete the sentence fragment.
Reviewer 2 Report
The manuscript titled “What Is the Best Approach for Identifying Missed Invasive Fungal Infections in Formalin-Fixed, Paraffin-Embedded Autopsy Specimens? An Autopsy Case of Disseminated Mucormycosis Developed after Allograft Hematopoietic Stem Cell Transplantation” is a work that can positively contribute to the field. I found no major revisions. Please change the title and make it shorter. The text is well written, except Table 2, which is full of typos; please revise.
Author Response
Thank you for your comment and their valuable suggestions. We updated the manuscript.
Comment from Reviewer 2:
The manuscript titled “What Is the Best Approach for Identifying Missed Invasive Fungal Infections in Formalin-Fixed, Paraffin-Embedded Autopsy Specimens? An Autopsy Case of Disseminated Mucormycosis Developed after Allograft Hematopoietic Stem Cell Transplantation” is a work that can positively contribute to the field. I found no major revisions. Please change the title and make it shorter. The text is well written, except Table 2, which is full of typos; please revise.
Our response:
In the point of Reviewers advise we changed the title more simply. “Comparison Approach for Identifying Missed Invasive Fungal Infections in Formalin-Fixed, Paraffin-Embedded Autopsy Specimens.”
Thank you for your suggestion. We revised the typos of Table 2.